# Cobalt-catalyzed electrooxidative C-H/N-H [4+2] annulation with ethylene or ethyne

Shan Tang[1], Dan Wang[2], Yichang Liu[2], Li Zeng[2] & Aiwen Lei[1,2,3]

Ethylene and ethyne are among the simplest two-carbon building blocks. However, quite limited methods can be applied to incorporate ethylene or ethyne into fine chemicals. Here we demonstrate a cobalt-catalyzed dehydrogenative C–H/N–H [4+2] annulation of aryl/vinyl amides with ethylene or ethyne by using an electrochemical reaction protocol. Significantly, this work shows an example of electrochemical recycling of cobalt catalyst in oxidative C–H functionalization reactions, avoiding the use of external chemical oxidants and co-oxidants. The electrochemical method provides a reliable and safe way for incorporating gas-phase ethylene or ethyne into fine chemicals. High reaction efficiency and good functional group tolerance are observed under divided electrolytic conditions.

---

[1] College of Chemistry and Molecular Sciences, Wuhan University, Wuhan 430072, China. [2] The Institute for Advanced Studies (IAS), Wuhan University, Wuhan 430072, China. [3] State Key Laboratory for Oxo Synthesis and Selective Oxidation, Lanzhou Institute of Chemical Physics, Chinese Academy of Sciences, Lanzhou 730000, China. Shan Tang and Dan Wang contributed equally to this work. Correspondence and requests for materials should be addressed to A.L. (email: aiwenlei@whu.edu.cn)

Ethylene is the most abundantly produced organic molecule by volume in petroleum industry[1,2]. A large amount of ethyne can be easily obtained by the partial combustion of methane[3]. Both ethylene and ethyne represent the simplest, readily functionalized, and inexpensive two-carbon synthons in organic chemistry. However, quite a limited number of methods have been reported to directly incorporate ethylene or ethyne into fine chemicals[4]. This might be partially due to the apprehensions about operating with gas-phase reactants, especially under oxidative conditions. Admittedly, the mixture of ethylene or ethyne with oxygen is quite dangerous since it can easily cause explosion. Thus, it is of great challenge to apply ethylene and ethyne as standard substrates in oxidative cross-coupling reactions.

Over the past decade, transition metal catalyzed oxidative C–H functionalization has been developed as an efficient and atom-economical method for the direct cross-coupling of arenes with alkenes and akynes[5–7]. However, most of these methods are only able to apply substituted alkenes or alkynes as the substrates[8]. Moreover, external chemical oxidants are generally required for the recycling of the metal catalysts, which inevitably brings undesired oxidation byproducts. In 2007, Jutand and co-workers[9] described an electrochemical oxidation promoted dehydrogenative Heck reaction between N-acetylanilines and alkenes. Pd(OAc)$_2$ was used as the key catalyst while 1,4-benzoquinone was added as a co-catalyst. During the reaction process, 1,4-benzoquinone acted as a redox mediator for oxidizing Pd(0) to Pd(II) by anode. Though with a limited substrate scope and low reaction efficiency, this work demonstrated the first example of electrochemical recycling of palladium catalyst in oxidative C–H functionalization reactions. During the following years, electrochemical oxidation has also been used in other palladium-catalyzed oxidative C–H functionalization reactions[10–13]. However, the electrochemical recycling of other transition metal catalysts in oxidative C–H functionalization has rarely been developed[14,15].

Owing to the low costs and low toxicity of cobalt salts, cobalt-catalyzed C–H functionalization has attracted increasing attentions over the past few years[16–18]. As for oxidative C–H functionalization, the stoichiometric amount of high-valent metal salts especially silver salts and manganese salts are generally required to recycle the cobalt catalysts[19–26]. In 2014, Daugulis and co-workers[27,28] first reported the cobalt-catalyzed oxidative C–H/N–H [4+2] annulation with alkynes and alkenes. In their reaction system, the stoichiometric amount of manganese salt was required for the reaction with alkyne and half an equivalent of manganese salt was essential for the reaction with alkene. More significantly, the use of oxygen as the terminal oxidant limits its potential application to incorporate ethylene and ethyne into fine chemicals due to the safety issues.

Here we demonstrate a cobalt-catalyzed electrooxidative C–H/N–H [4+2] annulation between aryl/vinyl amides and ethylene/ethyne. Cobalt catalyst is directly recycled by anodic oxidation under external oxidant-free conditions. During the submission of this manuscript, Ackermann and co-workers[29] reported a cobalt-catalyzed electrooxidative C–H/N–H [4+2] annulation between aryl/vinyl amides and terminal alkynes.

## Results

**Investigation of reaction conditions**. N-(quinolin-8-yl)benzamide (1a) was chosen as the model substrate to test the reaction conditions. After considerable efforts, dehydrogenative C–H/N–H [4+2] annulation with ethylene was successfully achieved by using Co(acac)$_2$ as the catalyst in the presence of sodium pivalate under divided electrolytic conditions. Eighty-nine percent isolated yield of the cyclization product **3aa** could be obtained at 4.0 mA constant current electrolysis after 4 h (Table 1,

**Table 1 Effects of the reaction parameters**

| Entry | Variation from the "standard conditions" | Yield (%) |
|---|---|---|
| 1 | None | 89 |
| 2 | CoCl$_2$ instead of Co(acac)$_2$ | 52 |
| 3 | Co(OAc)$_2$·4H$_2$O instead of Co(acac)$_2$ | 56 |
| 4[a] | without NaOPiv·H$_2$O | Trace |
| 5[a] | CF$_3$SO$_3$Na instead of NaOPiv·H$_2$O | Trace |
| 6[a] | HCOONa·H$_2$O instead of NaOPiv·H$_2$O | 16 |
| 7[a] | NaOAc instead of NaOPiv·H$_2$O | 72 |
| 8[a] | PhCOONa·H$_2$O instead of NaOPiv·H$_2$O | 85 |
| 9 | CH$_3$CN instead of CF$_3$CH$_2$OH | 21 |
| 10 | CH$_3$CH$_2$OH instead of CF$_3$CH$_2$OH | 35 |
| 11 | Platinum plate cathode | 83 |
| 12 | Carbon cloth cathode | 83 |
| 13 | Platinum plate anode | 42 |
| 14 | No electric current | n.d. |
| 15 | 8.0 mA, 2 h | 80 |
| 16 | 2.0 mA, 8 h | 82 |

Reaction conditions: **1a** (0.20 mmol), Co(acac)$_2$ (15 mol%), NaOPiv·H$_2$O (1.5 equiv), $^n$Bu$_4$NBF$_4$ (3.0 equiv), CF$_3$CH$_2$OH (8.0 mL) [anode], and NaOPiv·H$_2$O (1.5 equiv), HOPiv (10 equiv), MeOH (8.0 mL) [cathode] in and H-type divided cell with carbon cloth anode, nickel plate cathode and an AMI-7001-30 membrane, constant current = 4.0 mA ($J_{anode}$ = 1.8 mA/cm$^2$), ethylene balloon (1 atm), 70 °C, 4 h (3.0 F/mol). Isolated yields are shown. *n.d.* not dectected
[a]Changes are made only in the anode chamber

entry 1). It is worthy of noting that the aminoquinoline directing group was essential for this electrochemical reaction since no desired product could be observed by using pyridine or pyridine-N-oxide as the directing group. The choice of a cobalt catalyst precursor was important for obtaining a high reaction yield. Replacing Co(acac)$_2$ by CoCl$_2$ or Co(OAc)$_2$ led to decreased product yields (Table 1, entries 2 and 3). The additive was also beneficial for achieving a good reaction efficiency. Only trace amount of the desired product could be detected in the absence of NaOPiv·H$_2$O or using CF$_3$SO$_3$Na instead of NaOPiv·H$_2$O (Table 1, entries 4 and 5). Applying HCOONa·H$_2$O instead of NaOPiv·H$_2$O led to a low reaction yield (Table 1, entry 6). Seventy-two percent yield could be obtained with NaOAc (Table 1, entry 7) while 85% yield could be obtained with PhCOO-Na·H$_2$O (Table 1, entry 8). 2,2,2-Trifluoroethanol was found to be the most efficient solvent in this transformation while acetonitrile or ethanol gave dramatically decreased reaction yields (Table 1, entries 9 and 10). As for the selection of electrode material, a platinum plate cathode and a carbon cloth cathode showed similar reactivity with a nickel plate cathode (Table 1, entries 11 and 12). However, the platinum plate was not so efficient when it was used as the anode material instead of the carbon cloth anode (Table 1, entry 13). No desired product could be observed without electric current (Table 1, entry 14). Nevertheless, the operating electric current did not exert much influence on the reaction efficiency. Slightly decreased reaction yields could still be obtained either by increasing or decreasing the operating current (Table 1, entries 15 and 16). The result observed by increasing operating current provides possibilities for larger scale synthesis.

We have tried to do a 5 mmol scale reaction in a larger divided cell with 30 mA constant current. Carbon cloth was used as the electrode material for both anode and cathode. Delightfully, 0.90 g (66%) of **3a** could be obtained after 13 h electrolysis (Fig. 1).

**Substrate scope**. The scope of amides in this dehydrogenative annulation reaction was in turn explored (Fig. 2). Benzamides bearing one *ortho* methyl group or two *meta* methyl groups all showed similar reactivity with simple benzamide (**3aa-3ca**). Notably, bromide substituent at the *para* position was well tolerated, which provides possibility for further functionalization (**3da**). Amides bearing the electron-donating group or the electron-withdrawing group at the *para* position all showed good reaction efficiency in this transformation (**3ea** and **3fa**). Besides benzamides, other aromatic amides such as thiophene-2-carboxamide and furan-2-carboxamide were able to furnish the desired products in good yields (**3ga** and **3ha**). Significantly, vinyl amides were also suitable substrates in this electrooxidative [4+2] annulation reaction. Methacrylamide and cinnamamide could react with ethylene to afford the desired annulation products in moderate yields (**3ia** and **3ja**). Notably, tri-substituted vinyl amide could give the annulation product in 80% yield (**3ka**).

Under the similar reaction conditions as ethylene, ethyne was also found to be a suitable two carbon linker in this

dehydrogenative C–H/N–H [4+2] annulation (Fig. 3). Besides, similar functional group tolerance was observed for aryl amides (**3ab**). Benzamides bearing *ortho* and *meta* methyl groups showed good reactivity in this transformation (**3bb** and **3cb**). *Para*-bromo benzamide was still a suitable substrate though with a decreased reaction yield (**3db**). Both electron-rich and electron-deficient benzamides could afford the desired products in high yields (**3eb** and **3fb**). Thiophene-2-carboxamide and furan-2-carboxamide could also react with ethyne to give the desired aromatic compounds (**3gb** and **3hb**). Similar to the study with ethylene, vinyl amides were also tested as substrates. Methacrylamide reacted with ethyne with 49% yield (**3ib**) while cinnamamide could provide a 79% yield (**3jb**). Tri-substituted vinyl amide also furnished an annulation product in 76% yield (**3kb**). Besides the reaction with ethylene and ethyne, substituted alkenes and alkynes were also suitable in the electrochemical [4+2] annulation reaction. Selected examples are shown in Supplementary Fig. 3.

## Discussion

In the next step, experiments were carried out to get some insights into the reaction mechanism. First, kinetic isotope experiments were performed to probe the C–H activation step. An intermolecular competition experiment between **1a** and [**D₅**]-**1a** was done. The ratio of the [4+2] oxidative annulation products **3aa** and [**D₄**]-**3aa** was determined as 3.5:1 (Fig. 4a). Parallel reactions of **1a** and [**D₅**]-**1a** were carried out and the reaction rates were determined separately, giving a KIE value of 1.4 (Fig. 4b, $k_H/k_D$). Since only secondary isotope effect was observed in the parallel reactions, the C–H bond cleavage was a slow step but not likely to be the rate-determining step[30].

Preliminary kinetic studies were also carried out to determine the order of reaction components for the cobalt-catalyzed electrooxidative C–H/N–H [4+2] annulation between **1a** and ethylene. It is worthy of noting that induction period was observed in

**Fig. 1** Large-scale synthesis. Gram scale reaction with ethylene under electrochemical conditions

**Fig. 2** Electrooxidative C–H/N–H [4+2] annulation with ethylene. Reaction conditions: **1** (0.20 mmol), Co(acac)₂ (15 mol%), NaOPiv·H₂O (1.5 equiv), $^n$Bu₄NBF₄ (3.0 equiv), CF₃CH₂OH (8.0 mL) [anode], and NaOPiv·H₂O (1.5 equiv), HOPiv (10 equiv), MeOH (8.0 mL) [cathode] in and H-type divided cell with carbon cloth anode, nickel plate cathode, and an AMI-7001-30 membrane, constant current = 4.0 mA ($J_{anode}$ = 1.8 mA/cm²), ethylene balloon (1 atm), 70 °C, 4 h (3.0 F/mol). Isolated yields are shown

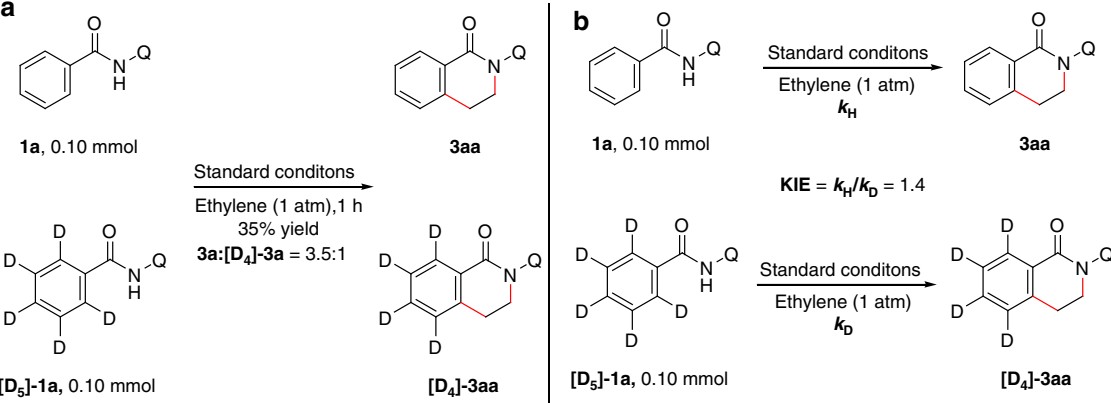

**Fig. 3** Electrooxidative C–H/N–H [4+2] annulation with ethyne. Reaction conditions: **1** (0.20 mmol), Co(acac)$_2$ (10 mol%), NaOPiv·H$_2$O (2 equiv), $^n$Bu$_4$NBF$_4$ (3.0 equiv), CF$_3$CH$_2$OH (8.0 mL) [anode], and NaOPiv·H$_2$O (2.0 equiv), HOPiv (10 equiv), MeOH (8.0 mL) [cathode] in and H-type divided cell with carbon cloth anode, nickel plate cathode, and an AMI-7001-30 membrane, constant current = 5.0 mA ($J_{anode}$ = 2.2 mA/cm$^2$), ethyne balloon (1 atm), 70 °C, 3 h (2.8 F/mol). Isolated yields are shown

**Fig. 4** Kinetic isotope effect experiments. **a** Intermolecular competition experiment between **1a** and **[D$_5$]-1a**. **b** Parallel reactions with **1a** and **[D$_5$]-1a**

most of the reactions. The initial reaction rates were monitored by GC analysis upon changing the concentration of **1a**, NaO-Piv·H$_2$O, and cobalt catalyst. The reaction rates were almost invariant under different concentrations of **1a** or NaOPiv·H$_2$O (Fig. 5a, b). Thus, substrate binding or deprotonation might not be involved in the rate-determining step. Interestingly, the reaction rate was even independent on the concentration of Co(acac)$_2$ (Fig. 5c) while the initial reaction rates changed with different operating currents (Fig. 5d). These results suggested that the electrochemical oxidation was likely to be the rate-limiting step during electrolysis.

After that, we studied the electrochemical oxidation step by performing cyclic voltammetry (CV) experiments. An obvious oxidation peak of substrate **1a** could be observed at 1.47 V (Fig. 6a, blue line). As for Co(acac)$_2$, two relatively weak oxidation peaks were observed at 1.12 and 1.53 V (Fig. 6a, red line). These results indicated that Co(II) species was probably oxidized

by anode before **1a**. Moreover, the mixture of Co(acac)$_2$ and **1a** demonstrated a stronger oxidation peak at 1.13V when compared with the oxidation peaks only observed with Co(acac)$_2$ (Fig. 6a, green line). According to our previous EXAFS study on the oxidation of Co(acac)$_2$ by Ag$_2$CO$_3$[31], Co(III) species were likely to be generated through the anodic oxidation of the coordinated Co(II) species. During current controlled electrolysis, the working voltage for whole electrolytic cell ranged from 7.50 to 8.50 V while the oxidation potential of anode (vs Ag/AgCl) ranged from 1.24 to 1.60 V. To confirm the assumption that coordinated Co(II) species were the key intermediate in electrochemical oxidation, the reaction was conducted at the first peak oxidation potential of the mixture of Co(acac)$_2$ and **1a**. Notably, the potential controlled electrolysis at 1.13 V afforded **3aa** in 70% yield (Fig. 6b).

Based on the above-mentioned results, a possible mechanism for this cobalt-catalyzed electrooxidative C–H/N–H annulation

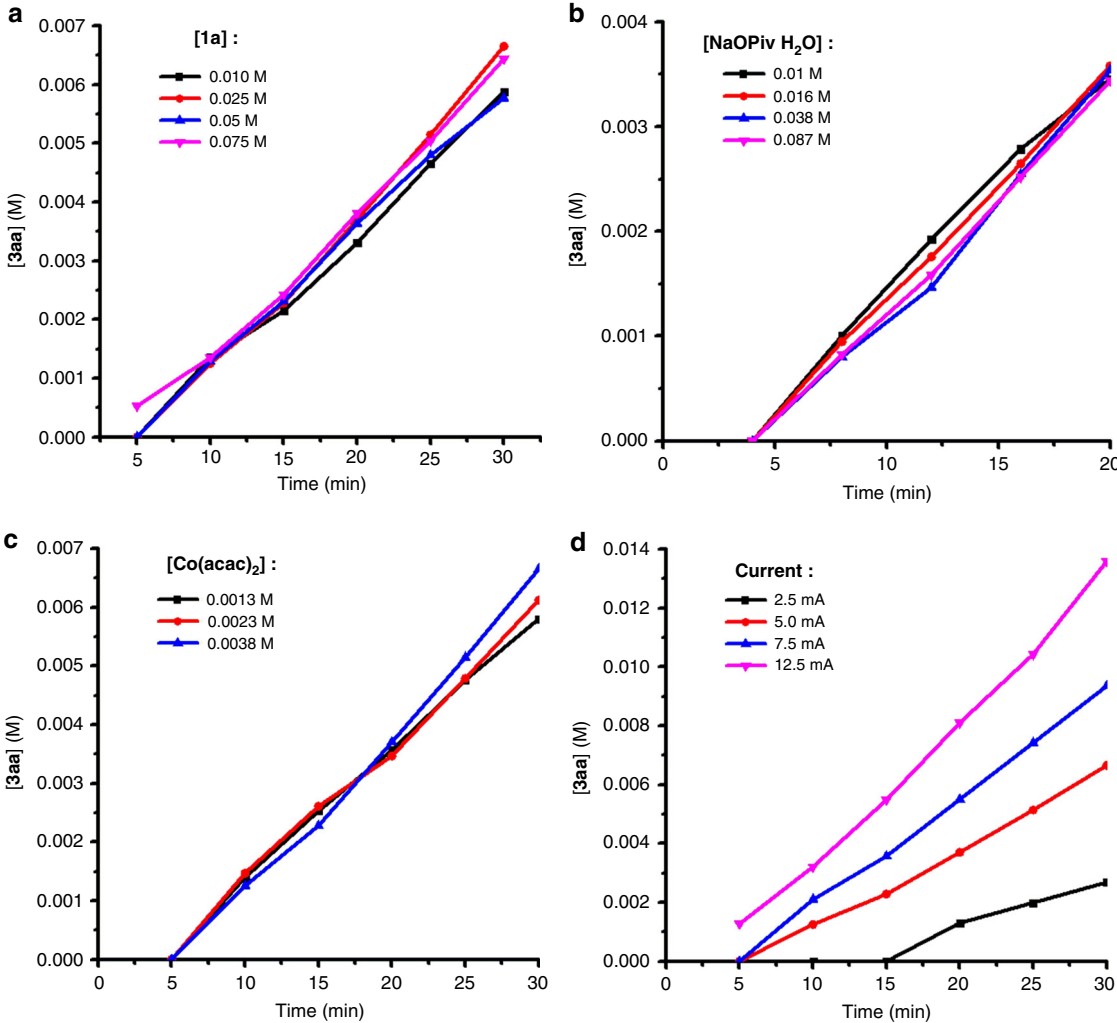

**Fig. 5** Kinetic studies for the cobalt-catalyzed electrooxidative C–H/N–H [4+2] annulation between 1a and ethylene. **a** Kinetic profiles under different concentrations of **1a**. **b** Kinetic profiles under different concentrations of NaOPiv·H$_2$O. **c** Kinetic profiles under different concentrations of Co(acac)$_2$. **d** Kinetic profiles under different operating currents

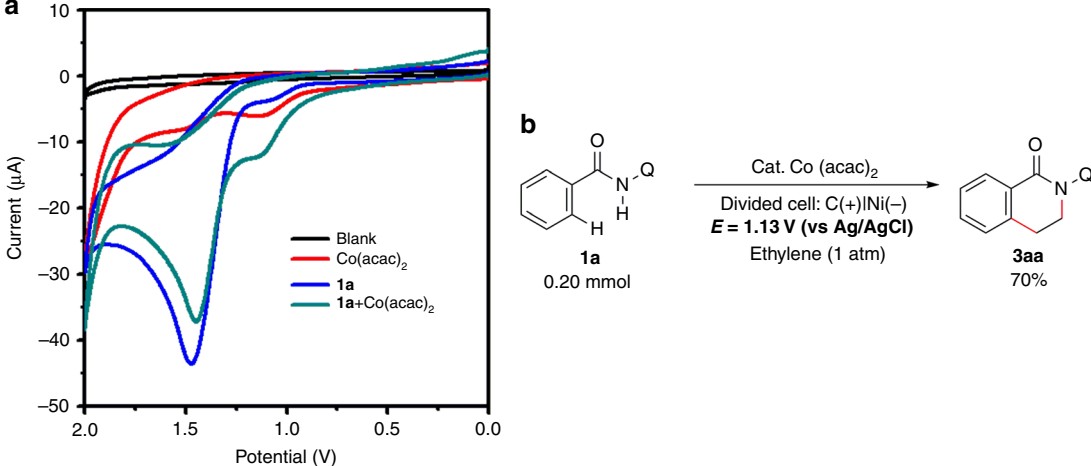

**Fig. 6** Study of the oxidation potential during the electrolysis. **a** Cyclic voltammograms in CF$_3$CH$_2$OH (10 mL) with 0.06 M $^n$Bu$_4$NBF$_4$: blue line, **1a** (0.050 mmol); red line, Co(acac)$_2$ (0.050 mmol); green line, a mixture of Co(acac)$_2$ (0.050 mmol), and **1a** (0.050 mmol). **b** Potential controlled electrolysis at 1.13 V (vs Ag/AgCl)

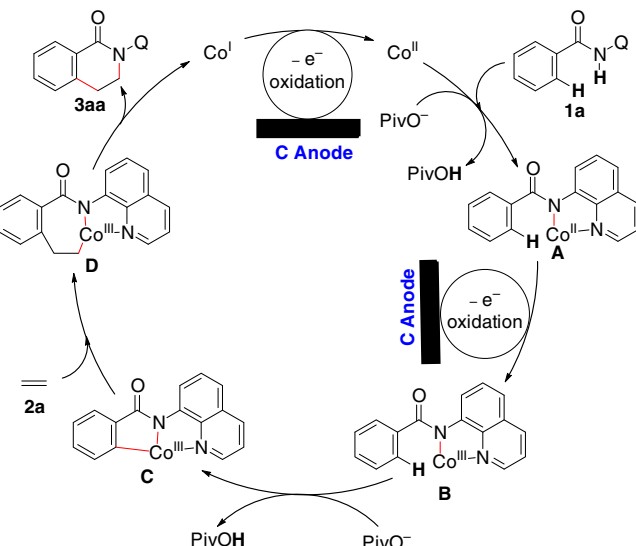

**Fig. 7** Proposed mechanism for the reaction between **1a** and **2a**. Tentative reaction mechanism involves anodic oxidation of the coordinated Co(II) complex to generate a Co(III) complex, intramolecular C−H activation, ethylene insertion, and reductive elimination of Co(III) species to form the annulation product, and finally a second anodic oxidation to regenerate the Co(II) catalyst

reaction between **1a** and ethylene is proposed in Fig. 7. Initially, Co(II) complex coordinates with **1a**, with the help of NaO-Piv·$H_2O$ affording a bidentate nitrogen coordinated Co(II) complex **A**. Next, complex **A** is directly oxidized by anode to afford Co(III) complex **B**. With the assistance of NaOPiv·$H_2O$, intramolecular C−H activation of complex **B** takes place to generate a cyclic Co(III) complex **C**. Then, ethylene (**2a**) insertion and reductive elimination of the Co(III) species form the final product **3aa**. Finally, the Co(I) species generated after reductive elimination are oxidized by carbon anode to regenerate the Co(II) catalyst. Proton reduction is likely to be the concomitant cathodic reaction since a large amount of hydrogen gas can be detected by GC in the reaction system after the reaction is stopped.

In conclusion, we have demonstrated an example of recycling the cobalt catalyst by electrochemical anodic oxidation in oxidative C–H functionalization reactions. This electrochemical reaction protocol enables the dehydrogenative C–H/N–H annulation with ethylene or ethyne under external oxidant-free conditions. The reaction exhibits high functional group tolerance. Both aryl and vinyl amides are suitable substrates in this transformation, which affords the desired [4+2] annulation products in good to high reaction yields. Preliminary mechanistic study indicates that electrochemical oxidation of coordinated Co(II) complex may be the key step during the reaction process. The study of recycling other transition metals by anodic oxidation in oxidative C–H functionalization reactions is underway in our laboratory.

## Methods

**Electrooxidative C-H/N-H [4+2] annulation with ethylene (Method A)**. The electrolysis was carried out in an oven-dried H-type divided cell equipped with two stir bars. Carbon cloth (15 mm × 15 mm × 0.36 mm) was used as the anode and the nickel plate (15 mm×15 mm×1.0 mm) was used as the cathode. The two electrodes were separated by an ULTREX® AMI-7001-30 membrane. The anodic chamber was added with amide (0.20 mmol), Co(acac)₂ (0.030 mmol, 7.7 mg), NaOPiv·$H_2O$ (0.30 mmol, 42.6 mg), and $^{n}Bu_4NBF_4$ (0.60 mmol, 197.6 mg) while the cathodic chamber was added with NaOPiv·$H_2O$ (0.30 mmol, 42.6 mg) and HOPiv (2.0 mmol, 204.3 mg). A balloon filled with ethylene (1 atm) was connected to the

electrolysis system and purged for three times. Subsequently, $CF_3CH_2OH$ (8.0 mL) and MeOH (8.0 mL) were added to the anodic chamber and cathodic chamber respectively. Then the electrolysis system was stirred at a constant current of 4.0 mA at 70 °C for 4 h. When the reaction was finished, the reaction mixture of the anodic chamber was washed with water and extracted with diethyl ether (10 mL × 3). The organic layers were combined, dried over $Na_2SO_4$, and concentrated. The pure product was obtained by flash column chromatography on silica gel (petroleum: ethyl acetate = 1:1). Full experimental details and characterization of the compounds are given in the Supplementary Information.

**Electrooxidative C-H/N-H [4+2] annulation with ethyne (Method B)**. The electrolysis was carried out in an oven-dried H-type divided cell equipped with two stir bars. Carbon cloth (15 mm×15 mm×0.36 mm) was used as the anode and the nickel plate (15 mm×15 mm×1.0 mm) was used as the cathode. The two electrodes were separated by an ULTREX® AMI-7001-30 membrane. The anodic chamber was added with amide (0.20 mmol), Co(acac)₂ (0.020 mmol, 5.1 mg), NaOPiv·$H_2O$ (0.40 mmol, 56.9 mg), and $^{n}Bu_4NBF_4$ (0.60 mmol, 197.6 mg) while the cathodic chamber was added with NaOPiv·$H_2O$ (0.40 mmol, 56.9 mg) and HOPiv (2.0 mmol, 204.3 mg). A balloon filled with ethyne (1 atm) was connected to the electrolysis system and purged for three times. Subsequently, $CF_3CH_2OH$ (8.0 mL) and MeOH (8.0 mL) were added to the anodic chamber and cathodic chamber, respectively. Then electrolysis system was stirred at a constant current of 5.0 mA at 70 °C for 3 h. When the reaction was finished, the reaction mixture of the anodic chamber was washed with water and extracted with diethyl ether (10 mL × 3). The organic layers were combined, dried over $Na_2SO_4$, and concentrated. The pure product was obtained by flash column chromatography on silica gel (petroleum: ethyl acetate = 1:1). Full experimental details and characterization of the compounds are given in the Supplementary Information.

**Data availability**. The authors declare that the data supporting the findings of this study are available within the article and its Supplementary Information files.

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

## Acknowledgements

This work was supported by the National Natural Science Foundation of China (21390402, 21520102003), the Ministry of Science and Technology of China (2012YQ120060), the Fundamental Research Funds for the Central Universities and the Program of Introducing Talents of Discipline to Universities of China (111 Program).

## Author contributions

A.L. and S.T. contributed to the conception and design of the experiments. S.T., D.W., and Y.L. performed the experiments. L.Z. and S.T. designed the H-type divided cell for this reaction system. S.T. and A.L. co-wrote the manuscript and all authors contributed to data analysis and scientific discussion.

## Additional information

**Competing interests:** The authors declare no competing financial interests.

