## [Peer Review File · Nature Communications]

Reviewer #1 (Remarks to the Author):

Lei and collaborators present here cobalt-catalyzed dehydrogenative C-H/N-H [4+2] annulation of aryl/vinyl amides with ethylene or ethyne by using an electrochemical reaction protocol. This protocol avoids the use of external chemical oxidants and co-oxidants, providing a green approach for C-H functionalization. However, a similar study has been reported by Lutz Ackermann (10.1002/anie.201712647) on the electrochemical alkyne annulation by cobalt-catalyzed C-H/N-H activation under ambient reaction conditions. In other words, this work does not show the first example of electrochemical recycling of cobalt catalyst in oxidative C-H functionalization reactions. In addition, the highly similar work has been demonstrated in Daugulis' group on cobalt-catalyzed oxidative C-H/N-H [4+2] annulation of aryl/vinyl amides with alkenes or alkynes (Angew. Chem. Int. Ed. 2014, 53, 10209-10212 and Org. Lett. 2014, 16, 4684-4687). For these reasons, I cannot recommend this publication of this work in Nature Communications.

Reviewer #2 (Remarks to the Author):

In their revised manuscript Lei and co-workers have performed a tremendous work to improve the experimental part to further substantiate the conclusions. In particular, I would like to emphasize the importance of the kinetic studies which now allow them to reveal the rate-controlling step of the proposed mechanism. Overall, I am very satisfied with their comments and I fully support the publication of this work in Nature Communications.

Reviewer #3 (Remarks to the Author):

The authors have undertaken significant additional experiments to address my previous comments and questions. The experiments on a wider range of proton acceptors, detection of hydrogen gas, alternative substrates, full cell potential, and mechanism help to elevate this work and further increase impact. Overall, I feel that the paper will be impactful due to high selectivity and electrochemical cycling of the catalyst, and that the work is suitable for publication in Nature Communications.

Response Letter

Reviewer #1 (Remarks to the Author):

Lei and collaborators present here cobalt-catalyzed dehydrogenative C-H/N-H [4+2] annulation of aryl/vinyl amides with ethylene or ethyne by using an electrochemical reaction protocol. This protocol avoids the use of external chemical oxidants and co-oxidants, providing a green approach for C-H functionalization. However, a similar study has been reported by Lutz Ackermann (10.1002/anie.201712647) on the electrochemical alkyne annulation by cobalt-catalyzed C-H/N-H activation under ambient reaction conditions. In other words, this work does not show the first example of electrochemical recycling of cobalt catalyst in oxidative C-H functionalization reactions. In addition, the highly similar work has been demonstrated in Daugulis' group on cobalt-catalyzed oxidative C-H/N-H [4+2] annulation of aryl/vinyl amides with alkenes or alkynes (Angew. Chem. Int. Ed. 2014, 53, 10209-10212 and Org. Lett. 2014, 16, 4684-4687). For these reasons, I cannot recommend this publication of this work in Nature Communications.

Our response: We have cited the similar work of Ackermann and co-workers published during the submission of our manuscript in Ref. 29. At the same time, the works of Daugulis have been clearly mentioned and a short comparison with our work has been made.

Reviewer #2 (Remarks to the Author):

In their revised manuscript Lei and co-workers have performed a tremendous work to improve the experimental part to further substantiate the conclusions. In particular, I would like to emphasize the importance of the kinetic studies which now allow them to reveal the rate-controlling step of the proposed mechanism. Overall, I am very satisfied with their comments and I fully support the publication of this work in Nature Communications.

Our response: Thanks for the comments.

Reviewer #3 (Remarks to the Author):

The authors have undertaken significant additional experiments to address my previous comments and questions. The experiments on a wider range of proton acceptors, detection of hydrogen gas, alternative substrates, full cell potential, and mechanism help to elevate this work and further increase impact. Overall, I feel that the paper will be impactful due to high selectivity and electrochemical cycling of the catalyst, and that the work is suitable for publication in Nature Communications.

Our response: Thanks for the comments.